# Reverse Thinking: A New Method from the Graph Perspective for Evaluating and Mitigating Regional Surface Heat Islands

**Zhaowu Yu** [1,*] , **Jinguang Zhang** [2], **Gaoyuan Yang** [2] **and Juliana Schlaberg** [3]

1    Department of Environmental Science and Engineering, Fudan University, Songhu Road 2005, Shanghai 200438, China
2    Department of Geosciences and Natural Resource Management, Faculty of Science, University of Copenhagen, Rolighedsvej 23, 1958 Copenhagen, Denmark; jizh@ign.ku.dk (J.Z.); gy@ign.ku.dk (G.Y.)
3    Faculty of Environmental Science, Technical University of Dresden, 01062 Dresden, Germany; juliana.schlaberg@mallbox.tu-dresden.de
*    Correspondence: zhaowu_yu@fudan.edu.cn

**Abstract:** Accurately locating key nodes and corridors of an urban heat island (UHI) is the basis for effectively mitigating a regional surface UHI. However, we still lack appropriate methods to describe it, especially considering the interaction between UHIs and the role of connectivity (network). Specifically, previous studies paid much attention to the raster and vector perspective—based on standard landscape configuration metrics that only provide an overall statistic over the entire study area without further indicating locations where different types of pattern and fragmentation occur. Therefore, by reverse thinking, here we attempt to propose a new method from the graph perspective which integrates morphological spatial pattern analysis (MSPA)—which is used to characterize binary patterns with emphasis on connections between their parts as measured at varying analysis scales, and habitat availability indices to evaluate and mitigate regional surface UHI. We selected the Pearl River Delta Metropolitan Region (PRDR), one of the most rapidly urbanized regions in the world as the case study (1995–2015). The results of the case study showed: (1) the core (UHI) type accounts for the vast majority of the MSPA model, with the relative land surface temperature (LST) rises, the proportion of the core type will increase, and it could influence the edge (UHI) type significantly; (2) the branch, bridge, and islet (UHI) types have similar results to the lower temperature (4 < Relative LST $\leq$ 6) area and account for the majority, indicating that these types are more susceptible to their surrounding environment; (3) the importance and extreme importance area (node) from 1995 to 2015 have increased significantly and mainly distributed in the urbanized areas, which means cooling measures need to be implemented in these areas in order of priority. Shifting the research logic of UHI evaluation and mitigation from "patch" to "network", we hold the point that the method (reverse thinking) has significant theoretical and practical implications for mitigating regional UHI and urban climate-resilience.

**Keywords:** urban heat island; graph theory; morphological spatial pattern analysis; habitat availability indices; land surface temperature; climate adaption

## 1. Introduction

The urban heat island (UHI) effect is widely acknowledged as one of the most serious urban environmental problems caused by urbanization, as well as an important factor hindering the sustainable development of cities [1–3]. Urbanization has changed land cover and use, induced changes in the surface materials (and colors) of buildings in urban areas, and the emission of anthropogenic heat has enhanced a UHI effect [4–8]. Generally, UHI can be classified into two categories: atmospheric UHI (AUHI) and surface UHI (SUHI) [1]. However, owing to the feasibility of consistent and repeatable observations of the remote-sensing-based LST, UHI effects can be measured from an extensive spatial perspective [9–11]. Therefore, LST is widely used to investigate the spatiotemporal patterns

of the SUHI and the associated thermal environment consequences, which are also a focus of this study. UHI has also caused negative consequences such as impaired air quality, increased energy and water consumption, and the impaired health and well-being of urban residents [12–14]. In addition, many studies have also shown that in the context of global climate change, the UHI and its adverse effects will be more serious [15–17].

There are many definitions (models) and corresponding studies to evaluate the UHI (and its intensity) in the past [2,9,18–21]. For instance, the widely used UHI definition is the temperature difference between urban and surrounding rural areas [1,22]. This classification has given researchers and decision-makers a simple and intuitive framework to separate the urban and rural effects on local climate, but it recently received critical challenges [1,18,23].

Researchers like Stewart and Oke [20] and Montgomery [24] pointed out that urban and rural areas are a dynamic and continuous process rather than a dichotomy; hence, the traditional urban–rural classification cannot explain the true status of cities, especially in cities in densely populated developing countries [25]. Therefore, Stewart and Oke [20] proposed a new schema of the local climate zone (LCZ) and believed that this classification can better evaluate the UHI intensity and model. The LCZ model includes 17 standard categories, and the UHI can be defined as the temperature difference between the compact high-rise building category and the low plant category ($\Delta$TLCZ 1−LCZ D). However, it remains to be noted that these definitions of UHI and corresponding studies are based on the simple patch-mosaic model concept (which exploits the overlay of different thematic layers), thus it is difficult to describe the overall pattern and connectivity of the UHI effect, especially in the regional (urban agglomeration) scale. For instance, Zhou et al. [26] found that increased shape complexity and variability of buildings and paved surfaces lead to an increase in surface UHI. Sun et al. [27] demonstrated that land surface temperature (LST) is determined by the heat capacity and exchange among different landscape patches, and spatial connectivity between the heat source and sink is the essential factor affecting the thermal flows between landscape patches. Other studies have further revealed that the increased regional connectivity of UHI patches enhances UHI intensity [25,28], but related studies have not received enough attention, as well as the potential role of the connectivity on regional UHI effect [27,29]. More importantly, though LST has been used as a proxy of UHI or UHI as a measure of LST differences between urban and non-urban areas as we stated above [9], we still lack an appropriate (morphological) method to describe the UHI pattern and network. This makes it difficult for us to accurately locate the spatial pattern and network of UHI on a regional scale, as well as to formulate effective mitigation and adaption strategies.

In addition, many studies have demonstrated that landscape composition and spatial configuration (or arrangement) can also influence the surface UHI effect [9,18,30]. Specifically, previous studies have suggested that the relative location of blue and green space in an urban area is important to alleviate the UHI effect [7,31]. Zhang et al. [32] also demonstrated that clustered green space in Phoenix enhances local cooling because of the agglomeration effect, while dispersed patterns of green space lead to greater overall regional cooling. However, most of the current UHI studies and corresponding landscape indices are raster and vector-based [29,33,34]; however, an appropriate method to describe the UHI pattern from a perspective of the graph is still lacking, which makes it difficult for planners and decision-makers to identify the key nodes and links of UHI patches to mitigate UHI effect effectively. Therefore, as a reverse thinking process, the connectivity metrics based on graphical representation is a significant step in effectively determining UHI networks (key nodes and links); hence, it is reasonable to suggest that if we blocked (or destroyed) the key nodes and links of the network, the regional heat island can be alleviated as the role of connectivity on UHI intensity has been confirmed.

On the other hand, it is clear that with the development of urbanization, the distance between cities tends to decrease and even disappear as urban land expansion accelerates (individual cities restricted by administrative boundaries being gradually merged); then

urban agglomeration, representing a group of cities with a compact spatial organization and close economic ties, have been formed [28,35]. During the urbanization and urban agglomeration development, the UHI effect transcended far from its physical boundary due to the loss of unaffected land in presumed suburban and rural areas [28,36,37]. Hence, studies have recognized that regional thermal environmental problems cannot be solved from a single city or UHI perspective, so the region heat island (RHI) was proposed by Yu, Yao, Yang, Wang and Vejre [25] to describe the thermal environment in an urban agglomeration context. Thus, it is clear that a thorough understanding of the UHI intensity pattern and network at the regional scale is critical to develop effective mitigation and adaptation strategies, as well as for regional sustainability [38–40].

Therefore, in this study, to address the insufficiencies of the regional UHI studies (from raster and vector to graph), we introduce a new method of integrating morphological spatial pattern analysis (MSPA) [41] and habitat availability indices [42,43] to map regional UHI network pattern. It can improve the UHI analysis from previous logic—based on standard landscape configuration metrics that only provide an overall statistic over the entire study area without further indicating locations where different types of pattern and fragmentation occur. To verify the validity of the method, the Pearl River Delta Metropolitan Region (PRDR)—one of the most rapidly urbanized regions in the world—was selected as the case (1995–2015) to answer the following questions: (1) is the new method of integrating MSPA and habitat availability indices applicable for evaluating surface UHI patterns? (2) What are the change and critical patches (key nodes and links) in regional surface UHI for PRDR from 1995 to 2015 based on the new method? (3) What are the contributions of the method to regional surface UHI mitigation in theory and practice? This study may provide a new promising method (angle) for UHI research, and more scientifically propose surface UHI mitigation strategies in the (agglomeration) region.

## 2. Methodology

### 2.1. MSPA Model

MSPA is a concept derived from mathematical morphology [41], which is a method for detecting image pixel patterns in the landscape and automatically classifies the pixel data of the focus feature class into a new structural connectivity feature class [44]. The MSPA depends on a single parameter only and can be used for characterizing binary patterns with emphasis on connections between their parts as measured at varying analysis scales [41,45]. It allows automatic classification by (raster) pixel and description of the geometry, pattern, and connectivity of the landscape, which improves the previous standard landscape-configuration-metrics-based pattern analysis [42]. Moreover, it can detect the connection structure reliably, which is a key feature to quantify the importance of individual landscape map elements in network analysis [42,43]. The seven basic pattern classes (core, bridge, loop, branch, edge, perforation, and islet) provided by MSPA are as followed, with a single edge width parameter (s) governing the entire classification process [41] (Figure 1 and Table 1).

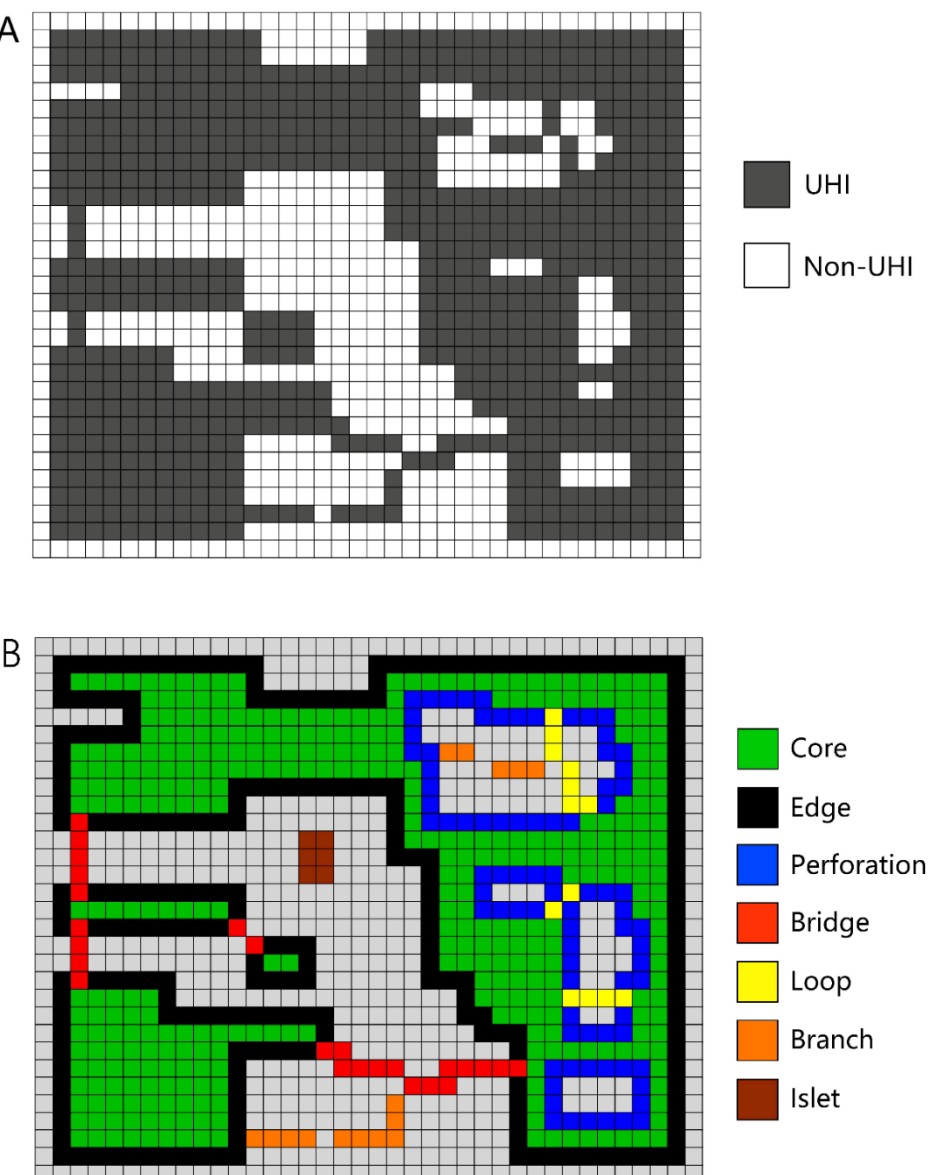

**Figure 1.** Illustration of the MSPA image classification of foreground pixels and legend. (**A**)The image background pixel color is grey and non-UHI, the black pixel is UHI. (**B**) show the different types of "UHI" in the context of the MSPA classification.

**Table 1.** Definition of morphological spatial pattern analysis (MSPA) classes and its meaning in the surface urban heat island (UHI) context.

| Class | Definition | Meaning in Surface UHI Context |
|---|---|---|
| **Core** | Foreground pixels surrounded on all sides by foreground pixels and greater than the specified edge width distance from the background. | *Core* is defined as those UHI pixels whose distance to the non-UHI areas is greater than the given edge width. |
| **Bridge** | Foreground pixels that connect two or more disjunct areas of the core. | *Bridge* defined as sets of contiguous non-core UHI pixels that connect at least two different core areas at their ends. They correspond to structural connectors or corridors that link different UHI core areas. |
| **Islet** | Foreground pixels that do not contain a core. The islet is the only unconnected class. | *Islet* defined as the isolated UHI patches that are too small to contain core pixels. |
| **Loop** | Foreground pixels that connect an area of the core to itself. | *Loop* similar to bridges but with the ends of the element connecting to different parts of the same core (UHI) area. |
| **Edge** | Pixels that form the transition zone between the foreground and background. | *Edge* defined a set of UHI pixels whose distance to the patch edge is lower than or equal to the given edge width and corresponds to the outer boundary of a core area. |

**Table 1.** *Cont.*

| Class | Definition | Meaning in Surface UHI Context |
|---|---|---|
| **Perforation** | Pixels that form the transition zone between the foreground and background for the interior regions of the foreground. The pixels forming the inner edge would be classified as perforations, whereas those forming the outer edge would be classified as the edge. | *Perforation* similar to the edge but corresponding to the inner boundary of a core (UHI) area. |
| **Branch** | Foreground pixels that extend from an area of the core, but do not connect to another area of the core. | *Branch* defined as the pixels that do not correspond to any of the previous six categories. It typically corresponds to an elongated set of consecutive UHI pixels that emanate from a UHI area and that do not reach any other UHI area at the other end. |

MSPA has been used in some fields, such as green infrastructure assessment and network optimization [46,47], ecosystem connectivity assessment [44], and forest conservation planning and management [42]. It is also used by government authorities, such as the Department of Agriculture Forest Service, United States and the European Commission [46]. However, according to the literature review, the use of MSPA was not found in the surface UHI study. Furthermore, it has been recognized that surface UHI can be regarded as a "thermal" landscape [25], then MSPA can be used to segment a raster "thermal" landscape binary map (i.e., UHI vs. non-UHI area) into different and mutually exclusive landscape pattern categories (Table 1). According to the advantages of the MSPA model and the "thermal" landscape characteristics, MSPA has the potential to be a promising model to evaluate the regional surface UHI pattern, and especially reveal the role of the connections (links).

In addition, MSPA classifications can be performed using 4 or 8 neighborhood rules. In this study, we used the default 8 neighborhood rule, i.e., if two pixels of the same class share one of their sides or vertices, they belong to the same "thermal" landscape element.

## 2.2. Habitat Availability Indices

It has been acknowledged that the level of landscape connectivity can quantitatively characterize whether a landscape is beneficial to the migration of species within the source patch, and maintaining good connectivity is conducive to ecosystem stability and biodiversity conservation [34,48]. Graph theory-based integral index of connectivity (IIC) [49], probability of connectivity ($0 \leq PC \leq 1$) index and node importance [33] can reflect the connectivity of landscapes and the important values of each patch in the landscape to the landscape connectivity, which has become an important indicator to measure the landscape pattern and habit conservation [43]. On the other hand, in the context of the regional surface UHI effect, if we can block (or destroy) the level of "thermal landscape" connectivity, then the regional surface UHI effect can be alleviated as stated above. Therefore, in this study, we used the IIC, PC and d$I$ (%) indexes to assess the regional surface UHI connectivity level, giving by

$$IIC = \frac{\sum_{i=1}^{n} \sum_{j=1}^{n} \left( a_i a_j / 1 + nI_{ij} \right)}{A_L^2} \quad (1)$$

$$PC = \frac{\sum_{i=1}^{n} \sum_{j=1}^{n} a_i a_j p_{ij}^*}{A_L^2} \quad (2)$$

$$dI\ (\%) = \frac{I - I_{remove}}{I} \times 100\% \quad (3)$$

where $n$ is the total number of habitat patches in the landscape, $a_i$ and $a_j$ are the areas of the habitat patches and $n1_{ij}$ is the number of links in the shortest path (topological distance) between patches $i$ and $j$. A path is a route along with connected nodes in which no node is visited more than once. $p_{ij}^*$ is the maximum product probability of all possible paths between patches $i$ and $j$. Here, $0 \leq IIC \leq 1$, $IIC = 0$ means that there is no connection between patches, and $IIC = 1$ means that they are fully connected between patches. $I$ is the

index value when the landscape element is present in the landscape and $I_{remove}$ is the index value after the removal of that landscape element (e.g., after a certain UHI patch loss).

These indices are particularly suitable for evaluating the contribution of each node and link to maintaining network connectivity [33,42]. That is to say, these indices adequately quantify the impact of removing specific UHI patches or corridors on "thermal" landscape connectivity, which is critical for a regional surface UHI alleviation study.

### 2.3. Framework of the Method

Figure 2 shows the framework of the new method from the graph perspective integrated MSPA and habitat availability indices. The underlying logic of this method (reverse thinking process) is that only by building the regional surface UHI network (key node and link) can we take better measures to alleviate the regional surface UHI effect. Specifically, after combining these two parts, a connectivity analysis—conducted by Conefor 2.6 software, will be processed. Subsequently, we can get the overall connectivity and node importance of the regional surface UHI, then the map of critical areas (UHIs) for connectivity would be mapped. Finally, theoretically, if we take measures (i.e., green and blue space) to remove or destroy the key nodes, links, and step-stones, then the regional surface UHI can be accurately alleviated.

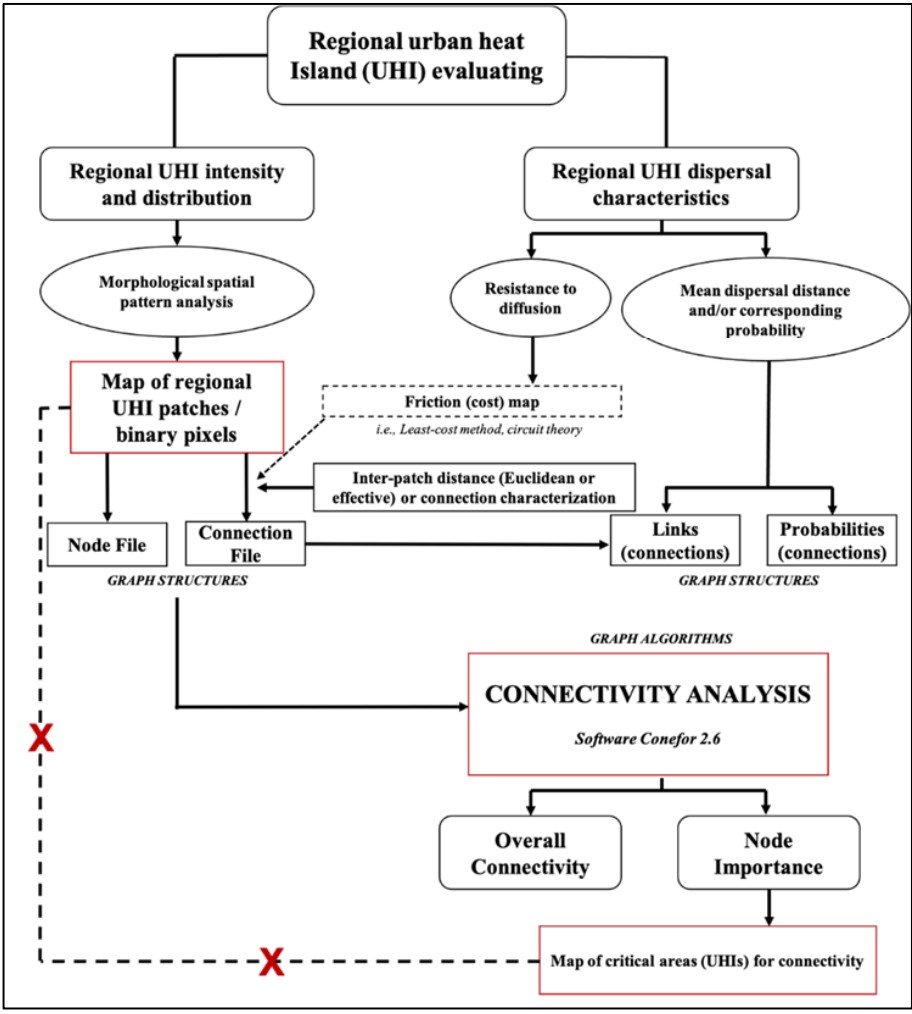

**Figure 2.** The framework for integrating MSPA and habitat availability indices for evaluating regional urban heat islands. Red "X" means block or destroy the connectivity of the result to alleviate the regional surface UHI effect.

## 3. Case Study

### 3.1. Study Area

This study uses the Pearl River Delta Region (PRDR) as an example to verify the effectiveness of this method. PRDR is located in southern China and is one of the three most developed and urbanized regions in China and the world (Figure 3). The total area of the PRDR is 55,900 km$^2$, including nine cities in Guangdong Province and two special administrative regions (SARs)—Hong Kong and Macau (Figure 3). Since the implementation of China's reform and opening-up policy in 1978, PRDR has been playing a leading role in China's economic and social development. At the same time, it has experienced rapid urbanization and large-scale urban population growth and land cover/land use change [50]. For example, as of the end of 2016, the PRDR's permanent population reached 68 million, and its GDP was USD 145.024 billion, accounting for 70% and 12% of the GDP of Guangdong Province and China, respectively (Statistics Bureau of Guangdong Province 2017). PRDR is a typical region of China's rapid urbanization also facing serious regional thermal environment problems [51,52]. Therefore, PRDR is an ideal area to verify the effectiveness of this method.

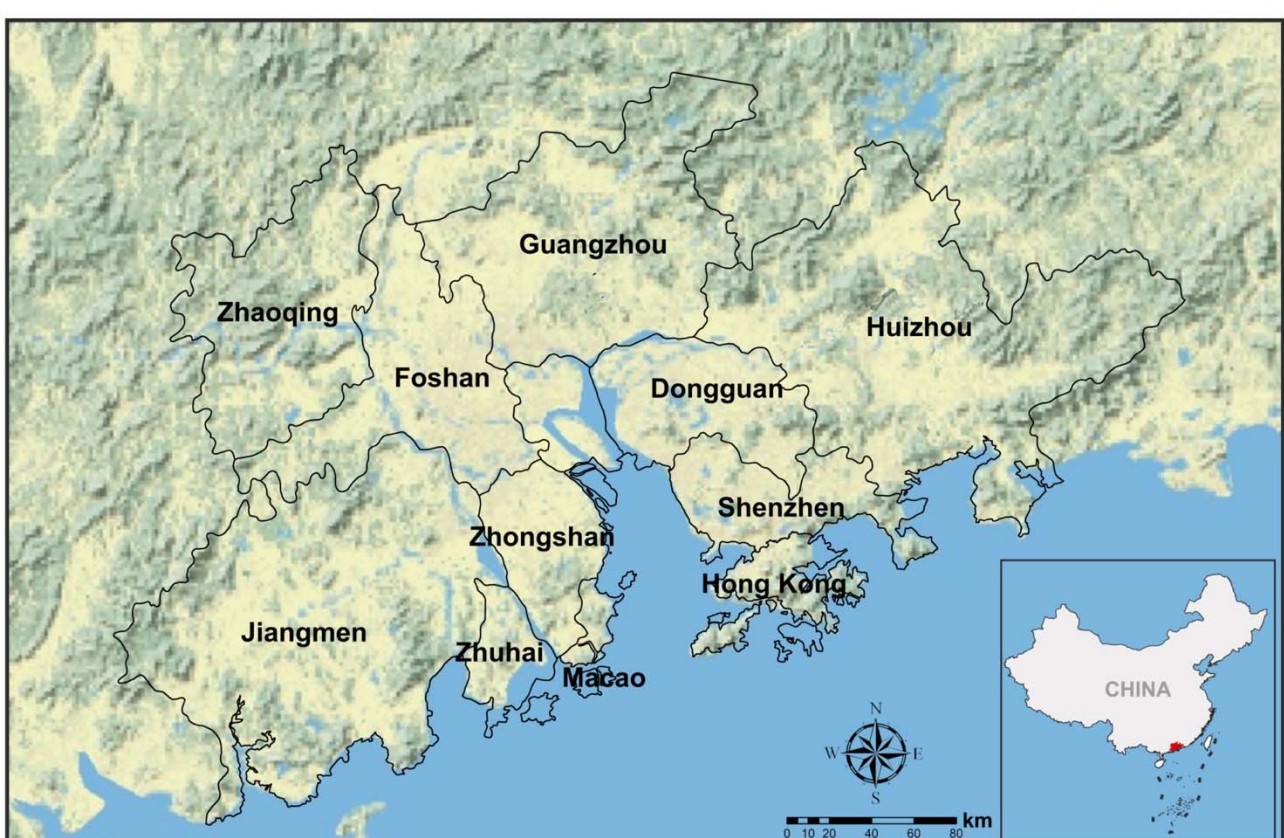

**Figure 3.** The study area (relative location of the Pearl River Delta Region (PRDR) in China and 11 cities in this region).

### 3.2. Data Preprocessing

#### 3.2.1. LST and Relative LST Calculation

In this study, five scenes of Landsat images were selected to cover the entire study area. At the same time, considering that the summer in the study area belongs to the rainy season, Landsat's image quality is usually poor during this period. Therefore, the data collection date was from June to September (1988–2017) (Table S1, Supplementary Materials). In addition, according to local historical climate records, the monthly temperature in June and September is 2 degrees Celsius lower than the average temperature in July and August,

so this error will be considered in further analysis. All Landsat images in this study were obtained from USGS Earth Explorer and processed in ENVI 5.3 software.

Previous studies have found that the LST retrieved by the radiative transfer equation algorithm can obtain the highest accuracy in the high atmospheric water vapor environments [7,10,53]. Therefore, this study chose the radiative transfer equation algorithm proposed by Jiménez-Muñoz et al. [54] to calculate LST. At the same time, to verify the accuracy of the LST results, we compared the corresponding temperature records obtained from the database of the Meteorological Service Database of Guangdong Province and found that the results were accurate and reliable (Table S2, Supplementary Materials). Finally, five LST scenes were retrieved separately for each year. We excluded areas contaminated by clouds or containing extreme values, and then merged the images. Subsequently, the relative land surface temperature (RLST) has been used as a reference value to compare LST differences between different years [55]. The RLST equation is:

$$RLST_j^i = LST_j^i - \overline{LST_j} \tag{4}$$

where $i$ represents each of the five years, $LST_j^i$ represents the remotely sensed LST of the pixels in year $j$, and $\overline{LST_j}$ represents the average LST of the whole area.

According to the results of the pilot studies in this region, the area of RLST > 4 °C is the stable and high-risk area from 1995 to 2015 and has been defined as the surface UHI [25,51]. Therefore, in this study, we defined the RLST > 4 °C as the surface UHI patch (<4 °C is the non-UHI patch). To track the surface UHI change more clearly, we also divided the RLST into three classes (4 °C < RLST ≤ 6 °C, 6 °C < RLST ≤ 8 °C, and RLST > 8 °C).

### 3.2.2. Land Cover Mapping from 1995 to 2015

In this study, Google Earth Engine (GEE) was used to draw land cover maps for 1995, 2005, and 2015. GEE is a cloud computing platform developed by Google, which can quickly access a large number of geospatial data sets at the planetary scale [56]. Here, we used 400 polygons representing each land cover type as training Data, and then obtained five types of land cover: built-up, forest, water, grassland, and bare land (Figure 4). At the same time, to test the overall accuracy of the classification, we randomly select 100 points from Google Earth every year to verify the accuracy of the land cover data. The results show that the overall accuracy exceeded 80% (i.e., 82%, 85% and 92% in 2005, 2010 and 2015, respectively). In addition, the results of Kappa Statistics were all greater than 0.8, which indicates that the classification meets the requirements of the research. For the technical details of the land cover map, please see the work conducted by Yu, Yao, Yang, Wang and Vejre [51].

Subsequently, we used a land cover dynamic degree index [57] to capture the most dynamic area from 1995 to 2015 in PRDR as the enlarged (specific) area in the next analysis:

$$S = \left\{ \sum_{ij}^{n} (\Delta S_{i-j}/S_i) \right\} \times \left( \frac{1}{t} \right) \times 100\% \tag{5}$$

where $S_i$ is the area of land cover type $i$ at the beginning of the period, $\Delta S_{i-j}$ is the total area of land cover type $i$ converted into other types. $t$ is the study period, and $S$ is the land use dynamic degree in the period of $t$. Finally, we found that Zhongshan City was the most dynamic area from 1995 to 2015 in the PRDR (Figure S1, Supplementary Materials), so we used Zhongshan City as the enlarged (specific) area in the next section.

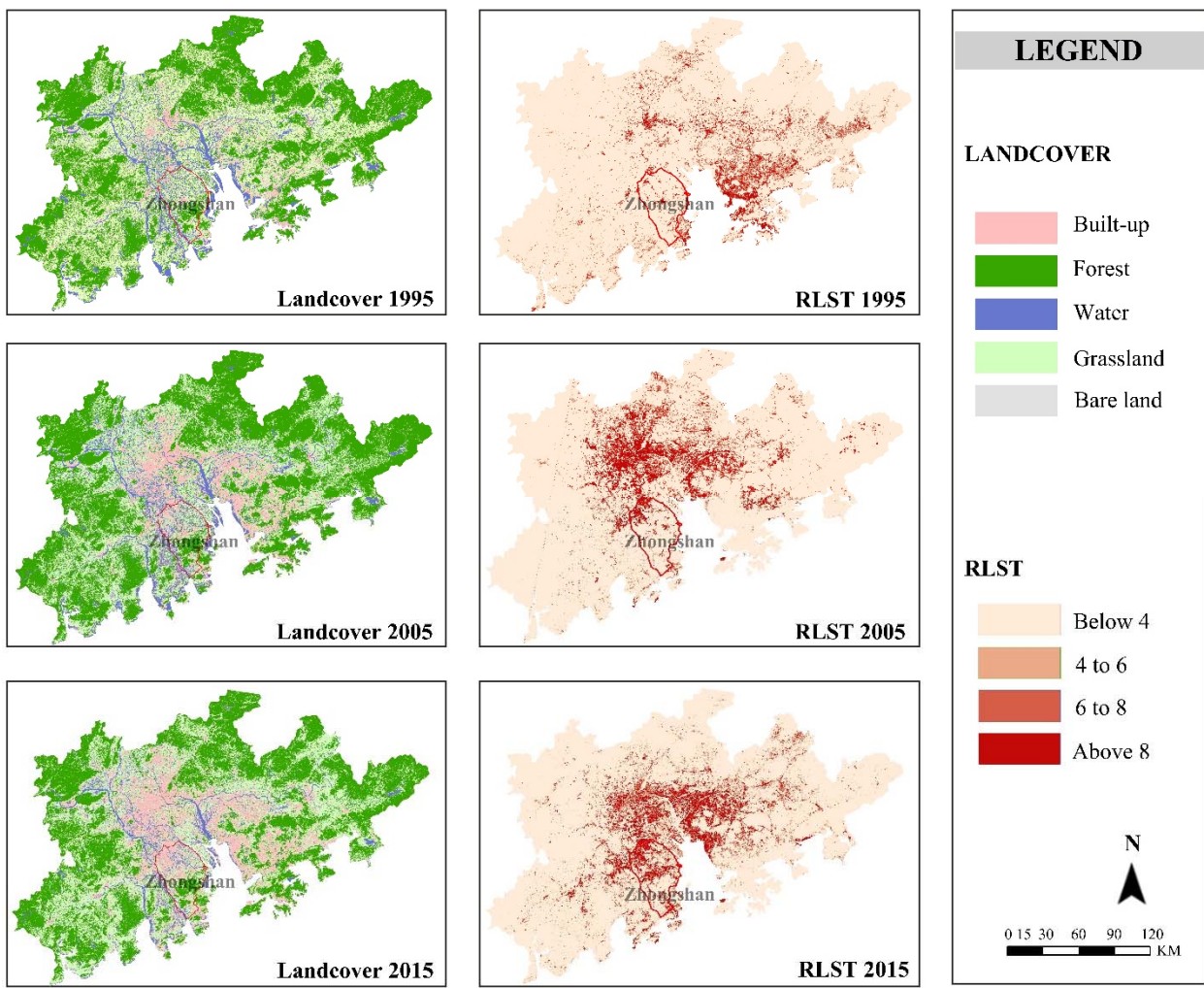

**Figure 4.** Land cover and the relative land surface temperature (RLST) maps of the PRDR from 1995 to 2015.

### 3.2.3. Analysis Process

In this case study, we first obtained the RLST and land cover map of PRDR from 1995 to 2015 and defined the RLST > 4°C as the surface UHI patch (foreground) based on the previous pilot study [25,51]. Then, we calculated the MSPA-based UHI pattern using the GuidosToolbox software. Second, we selected the *core* type as the source of the network for the next connectivity analysis (habitat availability indices). In this step, Conefor 2.6 software was used to calculate the connectivity of the UHI patch. Finally, we can obtain the key nodes (the importance of the key nodes will be ranked) and links of PRDR. Accordingly, we can evaluate the regional surface UHI pattern and can recommend accurate measures to alleviate it.

## 4. Results

### 4.1. Results of MSPA-Based Surface UHI Pattern Evaluation

The results of the MSPA-based surface UHI pattern from 1995 to 2015 are shown in Figure 5 and Table 2. It shows that the core type accounts for the vast majority of the MSPA model, and the areas of core type from 1995 to 2015 are 1582.06, 2738.26, 2166.46 km$^2$, respectively. Which account for the total area of surface UHI patches (and the total area of the region) 36.61% (3.84%), 42.42% (6.65%), and 29.66% (4.98%), respectively. In addition, the results also show that, as the RLST rises, the proportion (of the core type within the total area of surface UHI patches) will increase; and the class of RLST > 6 °C account for most of the core type (Table 2). As for the edge type, the results show that its proportion is

gradually decreasing (29.77%, 25.95%, and 22.46% respectively), and the RLST greater than 6 °C accounts for most of the edge type, which is similar to the core type. The reason for this pattern may be that the edge types generally surround the core types (Figure 1), so the core types would significantly affect the trend of edge types.

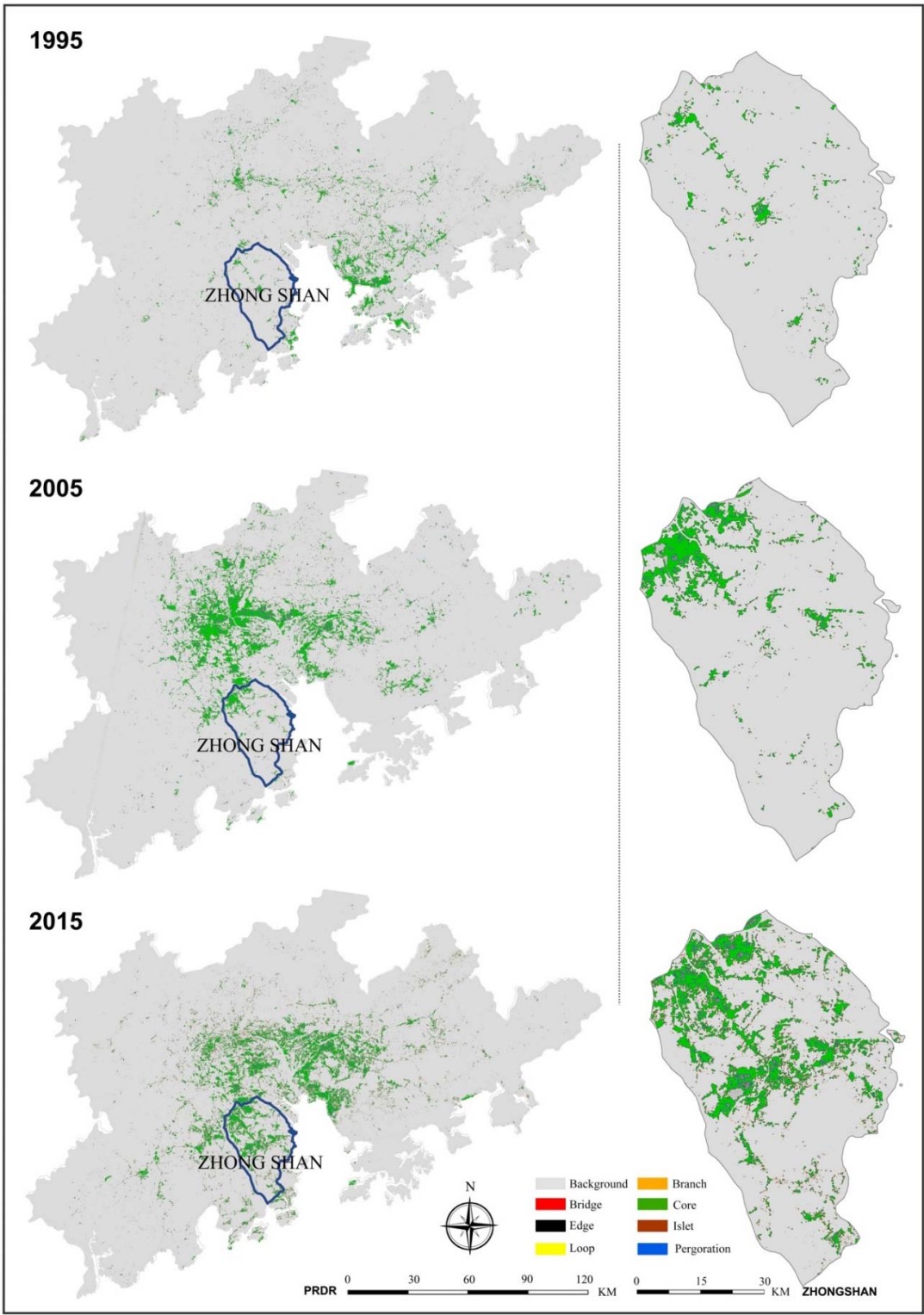

**Figure 5.** Results of the MSPA-based surface UHI patch classification from 1995 to 2015 in the PRDR and specifically for Zhongshan City (for detailed enlarged maps, please see the Supplementary Materials).

**Table 2.** MSPA-based classification and statistics results in different classes.

| Types | RLST Class | 1995 | | | 2005 | | | 2015 | | |
|---|---|---|---|---|---|---|---|---|---|---|
| | | Area (km²) | Accounting for the Total Area of UHI Patches (%) | Accounting for the Total Area (%) | Area (km²) | Accounting for the Total Area of UHI Patches (%) | Accounting for the Total Area (%) | Area (km²) | Accounting for the Total Area of UHI Patches (%) | Accounting for the Total Area (%) |
| Branch | $4 < RLST \leq 6$ | 335.38 | 6.70 | 0.81 | 359.06 | 5.56 | 0.871 | 618.91 | 8.48 | 1.50 |
| | $6 < RLST \leq 8$ | 207.87 | 4.15 | 0.50 | 234.56 | 3.63 | 0.569 | 395.52 | 5.42 | 0.96 |
| | >8 | 39.05 | 0.78 | 0.10 | 43.94 | 0.68 | 0.107 | 146.11 | 2.00 | 0.36 |
| Bridge | $4 < RLST \leq 6$ | 249.99 | 4.99 | 0.61 | 296.18 | 4.59 | 0.719 | 512.84 | 7.02 | 1.24 |
| | $6 < RLST \leq 8$ | 268.07 | 5.36 | 0.65 | 363.57 | 5.63 | 0.882 | 288.65 | 3.95 | 0.70 |
| | >8 | 6.46 | 0.13 | 0.02 | 9.21 | 0.14 | 0.022 | 43.25 | 0.59 | 0.11 |
| Core | $4 < RLST \leq 6$ | 307.24 | 6.14 | 0.75 | 271.06 | 4.20 | 0.658 | 356.54 | 4.88 | 0.87 |
| | $6 < RLST \leq 8$ | 535.42 | 10.70 | 1.20 | 590.76 | 9.15 | 1.434 | 168.97 | 2.31 | 0.41 |
| | >8 | 739.40 | 14.77 | 1.79 | 1876.44 | 29.07 | 4.553 | 1640.95 | 22.47 | 3.98 |
| Edge | $4 < RLST \leq 6$ | 460.70 | 9.20 | 1.12 | 423.63 | 6.56 | 1.028 | 608.05 | 8.33 | 1.48 |
| | $6 < RLST \leq 8$ | 653.82 | 13.06 | 1.59 | 703.27 | 10.89 | 1.707 | 314.61 | 4.31 | 0.76 |
| | >8 | 376.02 | 7.51 | 0.91 | 548.58 | 8.50 | 1.331 | 716.72 | 9.82 | 1.74 |
| Islet | $4 < RLST \leq 6$ | 451.47 | 9.02 | 1.10 | 370.65 | 5.74 | 0.899 | 573.53 | 7.85 | 1.39 |
| | $6 < RLST \leq 8$ | 151.49 | 3.03 | 0.37 | 105.65 | 1.64 | 0.256 | 517.09 | 7.08 | 1.26 |
| | >8 | 31.19 | 0.63 | 0.08 | 29.13 | 0.45 | 0.071 | 99.44 | 1.36 | 0.24 |
| Loop | $4 < RLST \leq 6$ | 93.33 | 1.86 | 0.23 | 87.22 | 1.35 | 0.212 | 125.89 | 1.72 | 0.31 |
| | $6 < RLST \leq 8$ | 72.76 | 1.45 | 0.18 | 68.34 | 1.06 | 0.166 | 84.77 | 1.16 | 0.21 |
| | >8 | 1.46 | 0.03 | 0.00 | 3.12 | 0.05 | 0.008 | 19.34 | 0.26 | 0.05 |
| Perforation | $4 < RLST \leq 6$ | 2.12 | 0.04 | 0.01 | 1.79 | 0.03 | 0.004 | 1.20 | 0.03 | 0.00 |
| | $6 < RLST \leq 8$ | 6.16 | 0.12 | 0.02 | 6.82 | 0.10 | 0.017 | 0.29 | 0.01 | 0.00 |
| | >8 | 16.36 | 0.33 | 0.04 | 62.99 | 0.98 | 0.153 | 69.16 | 0.95 | 0.17 |

Branch, bridge, and islet type have similar results, which shows that the lower temperature (4 < RLST ≤ 6) area accounts for the majority, indicating that these types are more susceptible to the ecological land's cooling effect (i.e., urban forest and water body). Such as in the branch type, the 4 < RLST ≤ 6 class account for 6.7% (the total of branch type is 11.63%), 9.89% (9.87%), 8.48% (15.9%), respectively. Furthermore, the results also show the increasing trend of the branch and bridge types within the total area of UHI patches, indicating that UHI patches are gradually expanding into the ecological land. For instance, the area of the bridge type increase from 10.48% (1995), 10.36%, 9.87% to 11.56% (2015). Generally, the loop type occupied a small part (3.34%, 2.46%, and 5.6% in 1995, 2005, and 2015, respectively) of the total area of UHI patches, and the lower temperature (4 < RLST ≤ 6) area account for the majority, the RLST greater than 8 °C can be ignored (i.e., the percentage of the RLST > 8 is 0.03% within the total area of UHI patches). The perforation type occupied the smallest area and proportions which can be almost ignored. In addition, as detailed for Zhongshan City, it also has similar proportions and trends.

Overall, the results of the MSPA-based surface UHI pattern evaluation from 1995 to 2015 show that the MSPA model can be a promising way to evaluate the surface UHI pattern and evolution due to it being able to reveal detailed information on the regional surface UHI pattern. Furthermore, these results also demonstrate that selecting the core type for further analysis is reliable.

### 4.2. Results of Surface UHI Connectivity Analysis

According to the results of the habit availability indices of *dA*, *dIIC*, and *dPC* in PRDR (Table 3) and the example city—Zhongshan (Table 4)—the core importance of UHI patches can be visualized shows in Figure 6. More detailed results can be seen in Supplementary Materials. We used the nature breaks (Jenks) which classified the UHI core importance into five grades (extreme importance, importance, general importance, less importance, and not importance) to accurately locate the areas with a severe UHI effect. For instance, in 2015, the areas that are of extreme importance, importance, general importance, less importance, and unimportance account for 55.07%, 29.15%, 10.30%, 4.25%, and 1.23% of the UHI patch in PRDR, respectively. In Zhongshan City, the percentage of five grades are 44.54%, 27.51%, 16.52%, 8.87%, 2.56% in 2015, respectively. Generally, from 1995 to 2015, the importance and extreme importance area of PRDR has increased significantly, which means that regional UHI effects are more and more severe and cooling measures must be implemented in these areas to mitigate the regional surface UHI effect. In addition, the results show that the key node areas are mainly distributed in the center of PRDR (delta area) where the urbanized and built-up areas are. Specifically, in 1995, the key node (extreme importance) areas were mainly located in Shenzhen city (and the surrounding area), which may explain that Shenzhen was undergoing large-scale industrialization and infrastructure construction during that period. For instance, the presence of a large amount of bare land in the construction process would cause a significant increase in LST [51]. In 2005, the extreme importance areas were mainly distributed in Guangzhou City, Foshan City, and Dongguan City, while this pattern changed in 2015 so that the key node areas were distributed in Dongguan city and Zhongshan city. Other cities in this region, like Jiangmen, Zhaoqing, and Zhuhai, have fewer key node areas from 1995 to 2015. Moreover, it needs to be mentioned that the distribution pattern of the key node area is relatively important, which means that although Shenzhen was an extremely important area in 1995, it does not mean that other areas (i.e., Guangzhou) are not important.

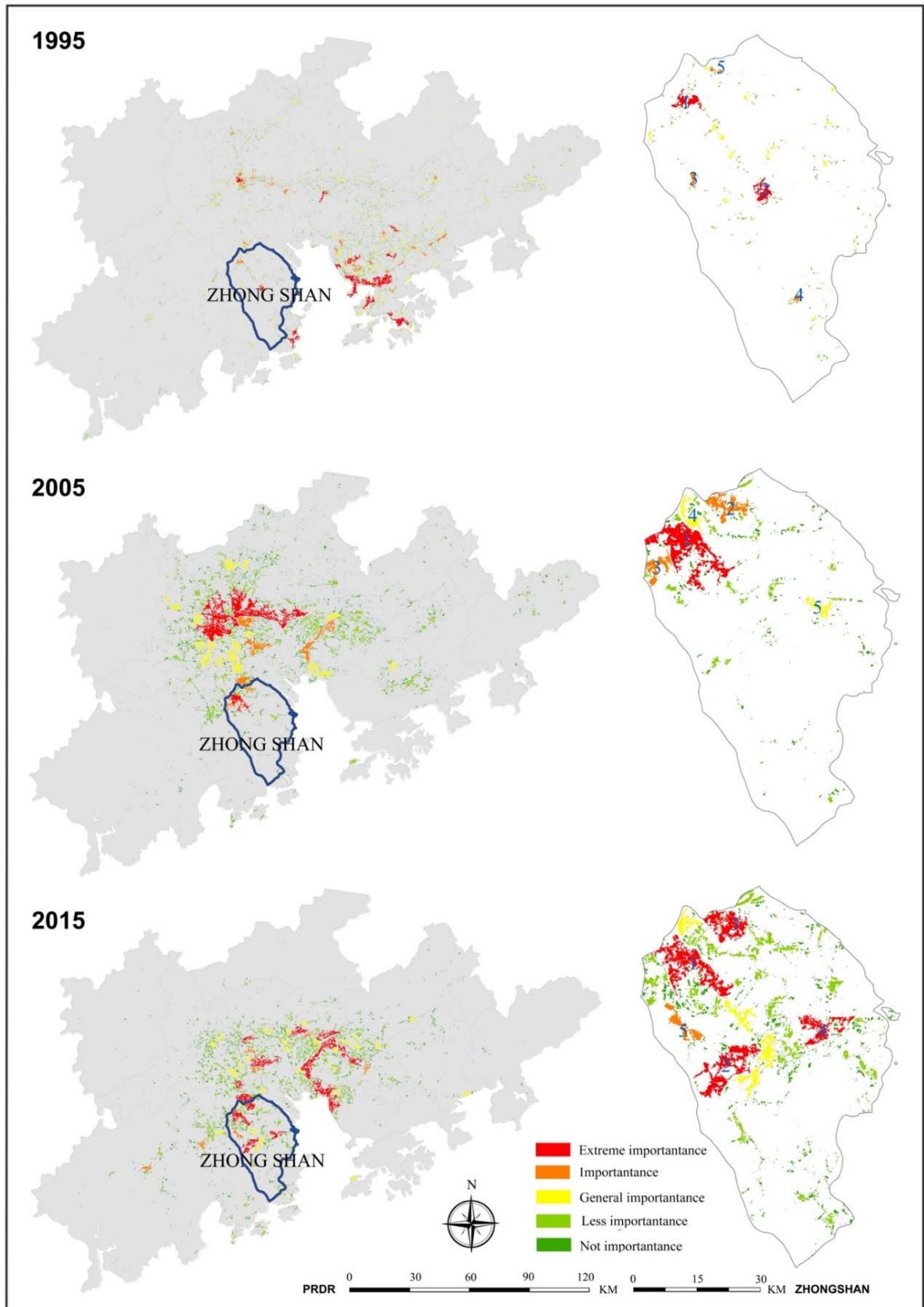

**Figure 6.** The results of core importance based on connectivity analysis from 1995 to 2015 in PRDR and specific Zhongshan City (detailed enlarged maps, please see the Supplementary Materials).

**Table 3.** Top 20 patch importance index ranking results from 1995 to 2015 in PRDR.

| Number | 1995 | | | 2005 | | | 2015 | | |
|---|---|---|---|---|---|---|---|---|---|
| | dA | dIIC | dPC | dA | dIIC | dPC | dA | dIIC | dPC |
| 1 | 16.93 | 35.91 | 37.41 | 14.26 | 29.57 | 29.94 | 5.11 | 10.66 | 11.28 |
| 2 | 4.11 | 8.10 | 7.92 | 13.33 | 27.04 | 27.65 | 3.03 | 6.40 | 6.56 |
| 3 | 2.86 | 6.02 | 6.60 | 2.93 | 6.20 | 6.64 | 2.66 | 5.84 | 6.02 |
| 4 | 2.24 | 3.64 | 3.47 | 2.86 | 5.32 | 5.18 | 2.31 | 4.63 | 4.84 |
| 5 | 1.47 | 2.28 | 2.02 | 2.29 | 4.83 | 5.39 | 2.15 | 4.51 | 4.78 |
| 6 | 1.38 | 2.59 | 2.56 | 2.07 | 3.96 | 4.06 | 1.93 | 3.92 | 3.91 |
| 7 | 1.30 | 2.79 | 3.05 | 1.92 | 3.55 | 3.54 | 1.77 | 3.38 | 3.30 |
| 8 | 1.23 | 2.67 | 3.01 | 1.83 | 3.71 | 3.74 | 1.65 | 3.34 | 3.42 |
| 9 | 1.22 | 2.63 | 2.96 | 1.36 | 2.53 | 2.48 | 1.41 | 2.93 | 3.02 |
| 10 | 1.09 | 1.85 | 1.79 | 1.22 | 2.48 | 2.62 | 1.31 | 2.75 | 2.91 |
| 11 | 1.08 | 1.82 | 1.70 | 1.10 | 2.10 | 2.02 | 1.12 | 2.196 | 2.22 |
| 12 | 1.02 | 2.15 | 2.19 | 1.10 | 1.91 | 1.88 | 0.97 | 1.97 | 1.98 |
| 13 | 0.97 | 2.05 | 2.26 | 1.10 | 2.19 | 2.25 | 0.87 | 1.73 | 1.81 |
| 14 | 0.92 | 1.86 | 1.95 | 1.10 | 1.91 | 1.88 | 0.74 | 1.50 | 1.55 |
| 15 | 0.80 | 1.40 | 1.40 | 1.08 | 2.12 | 2.23 | 0.73 | 1.51 | 1.63 |
| 16 | 0.78 | 1.66 | 1.78 | 0.91 | 1.90 | 2.09 | 0.70 | 1.28 | 1.16 |
| 17 | 0.68 | 1.18 | 1.07 | 0.91 | 1.82 | 1.96 | 0.66 | 1.43 | 1.51 |
| 18 | 0.68 | 1.48 | 1.54 | 0.90 | 1.65 | 1.62 | 0.64 | 1.26 | 1.28 |
| 19 | 0.65 | 0.99 | 0.89 | 0.86 | 1.72 | 1.70 | 0.61 | 0.91 | 0.68 |
| 20 | 0.64 | 1.16 | 1.08 | 0.75 | 1.50 | 1.56 | 0.61 | 1.19 | 1.14 |

**Table 4.** Top 10 patch importance index ranking results from 1995 to 2015 in Zhongshan city.

| Number | 1995 | | | 2005 | | | 2015 | | |
|---|---|---|---|---|---|---|---|---|---|
| | dA | dIIC | dPC | dA | dIIC | dPC | dA | dIIC | dPC |
| 1 | 18.21 | 38.64 | 41.36 | 39.33 | 78.79 | 81.71 | 18.21 | 38.64 | 41.36 |
| 2 | 9.68 | 23.89 | 22.60 | 11.83 | 14.21 | 10.96 | 9.68 | 23.89 | 22.60 |
| 3 | 9.05 | 28.05 | 22.05 | 7.64 | 7.59 | 6.56 | 9.05 | 28.05 | 22.05 |
| 4 | 4.91 | 2.88 | 4.07 | 4.37 | 2.26 | 1.27 | 4.91 | 2.88 | 4.07 |
| 5 | 3.46 | 2.33 | 2.38 | 2.90 | 2.51 | 4.48 | 3.46 | 2.33 | 2.38 |
| 6 | 2.63 | 4.52 | 2.86 | 2.17 | 1.71 | 1.68 | 2.63 | 4.52 | 2.86 |
| 7 | 1.97 | 6.20 | 2.40 | 1.11 | 1.03 | 0.83 | 1.9 | 6.20 | 2.40 |
| 8 | 1.80 | 2.22 | 1.26 | 1.09 | 0.10 | 0.12 | 1.80 | 2.22 | 1.26 |
| 9 | 1.64 | 3.13 | 2.38 | 1.08 | 1.01 | 0.41 | 1.64 | 3.13 | 2.38 |
| 10 | 1.41 | 0.35 | 0.71 | 1.04 | 0.85 | 0.42 | 1.41 | 0.35 | 0.71 |

Details for Zhongshan City (Figure 6 and Table 4), from the extremely important types of sporadic distribution in 1995 to the extremely important types, were gradually connected in 2005 and mainly concentrated in the northwest part, and by 2015, the extremely important types gradually became a network. Rankings of the core importance suggest to decision-makers and planners that corresponding surface UHI mitigation measures should be addressed. This result of Zhongshan city also clearly reflects the impact of urbanization and land-use change (Figure 4) on the regional UHI effect.

## 5. Discussion

### 5.1. Theoretical and Practical Implications

5.1.1. Theoretical Implication: From Patch to Network

It has been widely acknowledged that most surface UHI-related studies are raster and vector-based from a patch perspective [29,33], such as that of quantifying the threshold size and shape of a blue–green space to mitigate the UHI effect [19,31,58] and the optimal composition and configuration of blue–green space to achieve the best cooling effect [6,30,59]. However, the most important step is to accurately locate the most critical location and corridor of UHI patches before implementing these mitigation measures. Otherwise, the proposed UHI mitigation measures cannot be well integrated with urban climate adaption planning. Especially with the continuous expansion of urban areas and the emergence of urban agglomerations, traditional single UHIs have gradually emerged with multiple heat islands of different intensity [25,35]. More importantly, previous studies have shown

that at the scale of urban agglomerations, different surface UHI patches can transcend their physical (administrative) boundaries and influence each other, thereby exacerbating the regional heat island effect [28]. Therefore, an appropriate method to mapping the surface UHI pattern from a graph perspective is critical to mitigating a regional heat effect, particularly in an urban agglomeration scale.

In this study, through reversing thinking, we proposed a new method from the graph perspective, integrating morphological spatial pattern analysis and habitat availability indices to evaluate and mitigate regional heat islands. Theoretically, as shown in Figure 7, when we regard the regional heat island patches as nodes in the network (or as habitat patches in conservation biology), then different patches in the network are of different importance; and if these key nodes and corridors are destroyed, the regional thermal environment can be significantly alleviated. This thinking logic will elevate the related research from the patch to network perspective and will have an important impact on the evaluation and mitigation of regional heat islands. Therefore, in contrast to previous perspectives (patch-based), a more comprehensive perspective is that when we first consider the regional surface UHI evaluation and mitigation from the network (graph), the following step is to consider the effect of the patch (i.e., patch size, composition, and configuration). These two can be regarded as a unified closed loop starting from the graph. According to this theoretical framework, in general, we can first identify the critical UHI nodes and corridors, then mitigate these pinch points from the perspective of the patch.

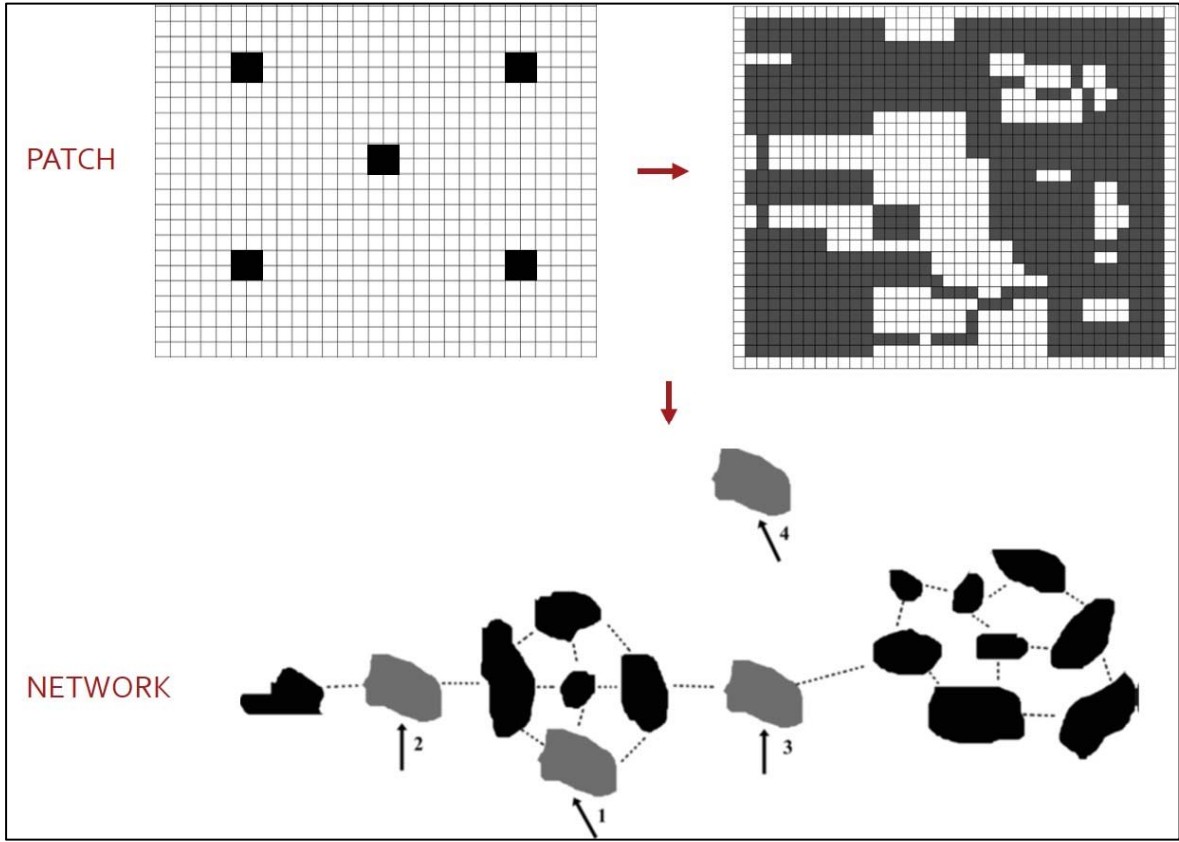

**Figure 7.** The conceptual diagram from the patch to network in surface UHI studies. The top left black square means the UHI patch (vector), and the top right is the same as the MSPA model (raster). Below is an example to illustrate different UHI patches in terms of their different connectivity roles (graph). Specifically, patch 1 is connected but not a key "stepping stone", but patches 2 and 3 are key "stepping stones" that need to be blocked or destroyed as a priority. Furthermore, patch 3 is more critical than patch 2 due to a more important connectivity role in a network. Patch 4 is an isolated UHI parch. Therefore, considering the connectivity role of the UHI patches in a network, Patch 3 > Patch 2 > Patch 1 > Patch 4, so these mean that patch 3 needs to be priority considered to block/destroy to mitigate regional heat island effect.

5.1.2. Practical Implication: Regional Surface UHI Mitigation

As stated above (in the introduction), many definitions and corresponding studies have been proposed to evaluate the UHI pattern, but these methods have their flaws [1,20]. In this study, from a graph perspective, we introduced the MSPA model to evaluate the regional surface UHI pattern, which can obtain detailed UHI information compared with the previous definitions. For example, in this PRDR case, we identified that the core type is becoming more and more important in the process of rapid urbanization, and it could influence the edge type significantly. Hence, this result implies that the cooling measures are crucial for core types to alleviate the expansion of the UHI effect. Which means that when doing the climate adaption planning, it is necessary to consider the construction of sufficient facilities such as green/blue spaces, green roofs, and cool materials in core type areas to cool down. The results of other types (such as branch and bridge) indicate that these heat patches (or we can call it corridors) are needed to be blocked/destroyed by urban blue–green space and green roof [6,58]. In addition, we quantified the core importance of UHI patches from 1995 to 2015 in PRDR and exampled Zhongshan City, which can provide a direct basis for decision-makers in PRDR to determine the priority of regional thermal environment mitigation and governance. This method is particularly useful in urban renewal based on climate mitigation and adaptation, that is, urban and regional heat island effects can be mitigated by firstly identifying key heat patches and corridors, then block/destroy them. Generally, from a graph perspective, this new method (reverse thinking) integrating MSPA and habitat availability indices can not only determine different UHI types more accurately and take corresponding mitigation measures effectively in the PRDR but can also be widely used in research and practice in other regions.

*5.2. Limitations and Further Study*

First, due to the research aims, this study only proposed a (reverse thinking) frame work—which is also the most important value of this research, and selected PRDR as the case to verify the validity of the method. However, the study did not provide a detailed method on how to build a surface UHI network, such as create resistance to diffusion or friction map (Figure 2). In the next step, it is necessary to identify the heat corridor through constructing the resistance surface [60], methods like circuit theory and the least-cost method can be employed. Second, the MSPA model is sensitive to the scale effect (i.e., the resolution of the raster image) [41], as well as the UHI determinants [10]. Hence, scale effects (grain size and spatial extent) must be considered in future related research. Third, the heat is in a state of continuous flow and might be affected by uncontrollable factors such as wind direction and wind speed. This is another limitation of this study and needs to be further considered when constructing the thermal network in the future. In addition, this study only used LST to do the analysis, but it is necessary to use air temperature to build the heat network in future study; for the reason that RS-based LST cannot capture the latent and sensible heat exchange [61], as well as the anthropogenic heat emission.

**6. Conclusions**

It is well recognized that it is difficult for many raster- and vector-based UHI studies to accurately locate key nodes and corridors on a regional scale, making the surface UHI mitigation measures difficult to be well integrated with urban climate adaption planning. In this study, from the graph perspective, we proposed a new method of integrating MSPA and habitat availability indices for evaluating and mitigating regional heat islands. Although these indices are not new in other research fields, this proposed method and framework is new in surface UHI studies. The underlying logic of this method (reverse thinking process) is that only by building the regional UHI network (key node and corridor) can we take better measures to alleviate the regional UHI effect. To verify the validity of the method, we selected PRDR (and gave the specific example of Zhongshan City) as the case. The case study found that (1) the core type accounts for the vast majority of the MSPA model from 1995 to 2015, and with the RLST rises, the proportion (of the core type within the total area

of UHI patches) increases, and it could influence the edge type significantly; (2) the branch, bridge, and islet type have similar results that the lower temperature ($4 < RLST \leq 6$) area accounts for the majority, indicating these types are more susceptible to ecological land's cooling effect. Moreover, the loop and perforation type occupied a small part with lower RLST and can be almost ignored; and (3) the importance and extreme importance area of PRDR have increased significantly and were mainly distributed in the urbanized and built-up area, which means cooling measures need to be implemented in these areas to mitigate the regional surface UHI effect.

Furthermore, due to the objectives of this research, we did not discuss in-depth how to build a heat island network and then destroy/block it. In subsequent steps, we suggest that methods like circuit theory and the least-cost method can be employed to build the heat network. Conclusively, the method and reverse thinking process are significant for evaluating and mitigating regional heat islands, and for urban (region) sustainable development.

**Supplementary Materials:** The following are available online at https://www.mdpi.com/2072-429 2/13/6/1127/s1, Table S1: Requisition date of the Landsat images in the preprocessing composite for retrieving LST.; Table S2: The minimum, maximum and mean LST for each year and the corresponding air temperature record obtained from the Meteorological Service Database of Guangdong Province (MSDGP). The mean LST for 2005 was the highest (36.18 °C), while that of 2010 (32.33 °C) was the lowest. The standard deviations of mean LST for 2005 (6.18 °C) and 2015 (6.25 °C) are higher when compared to other years (ranging from 2 to 4.62 °C). Moreover, comparing the mean LST to historical air temperatures (obtained from MSDGP), it was seen that their differences were no more than 10 degrees and the changing trend was similar. These results mean that the LST can, to some extent, represent the thermal environment dynamics of the Dongguan city (it needs to be noted that, at the same time, the LST is generally higher than the air temperature); Table S3: Node importances 1995–2015 Figure S1: Result of the land cover dynamic degree index from 1990. Figure S2: Top 20 nodes in 1995, 2005 and 2015.

**Author Contributions:** Conceptualization, Z.Y., and methodology, Z.Y., J.Z., and G.Y., processing and analyses of data: J.Z., visualization, J.Z., and G.Y., and J.S.; writing—original draft preparation, Z.Y., and G.Y.; writing—review and editing, Z.Y., G.Y.; supervision, Z.Y. and G.Y., All authors have read and agreed to the published version of the manuscript.

**Funding:** This research was funded by the Open Foundation of the State Key Laboratory of Urban and Regional Ecology of China (grant no. SKLURE2019-2-6).

**Institutional Review Board Statement:** Not applicable.

**Informed Consent Statement:** Not applicable.

**Data Availability Statement:** The data presented in this study are available on request from the corresponding author.

**Acknowledgments:** We also thanks anonymous reviewers and academic editor for their constructive comments and suggestions.

**Conflicts of Interest:** The authors declare no conflict of interest.

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
