# Peer review of "Reverse Thinking: A New Method from the Graph Perspective for Evaluating and Mitigating Regional Surface Heat Islands"

_remotesensing, doi:10.3390/rs13061127_

Round 1
Reviewer 1 Report
The paper #1125509 presents an interesting research able of shifting the point of view for evaluating and mitigating regional surface heat islands.
Each phase of the research is explained; the authors explain in more detail the theoretical aspects than those deriving from the case study. This makes the overall reading of the paper not fluid, as does the understanding of case study analysis.
In addition, the process images of the case study are small; the paper could be enriched with wider and clearer images, compared to the target of the research.
English language and style are fine, minor spell check is required; the abstract should be revised, to make it clearer.
Author Response
Answer: Thank you for your constructive comments!
We have updated the abstract and corrected the language error.
We also put clear maps as additional material. In Appendix file, the reader can see clear map of the research.
Reviewer 2 Report
In the article, theuthors describe a new methodology for evaluating and mitigating regional surface heat islands. The research subject matter raised in the work is important and worth mentioning. My main objections to the article can be described in the following points:
- On page 9 (line 288) the authors describe classification overall accuracy (OA). This seems to be an insufficient characteristic to describe the quality of land cover. The maps provided (Figure 4) should encourage the authors to describe the classification in more detail (e.g. with the Kappa Statistics)
- By taking into account in the RLST calculations the average of the entire study area, the method described is very dependent on the area studied. As a result, the RLST results for individual years (1995, 2005, 2015) are incomparable with each other, because they are related to a different value. Perhaps it is worth testing the RLST calculations against an overall multi-year mean?
- For the reviewer, the RLST values for Shenzhen are worth noting. Comparison between 1995 and 2015. One can see a significant decrease in RLST despite increasing built-up area. What can such a condition be caused by? Does it not challenge the overall analysis?
- Please add to the article (or additional materials) a table showing the calculations for formula (5) for individual parts of the PRDR
Other necessary fixes:
- Maps at work, in particular Figure 3, are of terrible quality and should be improved according to cartographic principles.
- In Figure 5, it should be clearly indicated whether the scale refers to the PRDR map or to the zoom.
- In all tables, consider the accuracy of the numbers. A notation of 0.000 in relation to the% value seems too accurate. In tables 3 and 4, the accuracy of the notation of values is significantly too high. Please round them properly.
Author Response
Reviewer 2
In the article, the authors describe a new methodology for evaluating and mitigating regional surface heat islands. The research subject matter raised in the work is important and worth mentioning.
My main objections to the article can be described in the following points:
- On page 9 (line 288) the authors describe classification overall accuracy (OA). This seems to be an insufficient characteristic to describe the quality of land cover. The maps provided (Figure 4) should encourage the authors to describe the classification in more detail (e.g. with the Kappa Statistics)
Answer: Thank you for your comments!
We have also used Kappa Statistics to see the overall accuracy. We can see the value of the Kappa Statistics are more than 0.8, which means that the classification is qualified to do such analysis.
- By taking into account in the RLST calculations the average of the entire study area, the method described is very dependent on the area studied. As a result, the RLST results for individual years (1995, 2005, 2015) are incomparable with each other, because they are related to a different value. Perhaps it is worth testing the RLST calculations against an overall multi-year mean?
Answer: Thank you for your comments!
Yes, as you mentioned, comparable is critical in this study, and actually, the RLST is used to do the comparison.
The RLST is calculated by each year’s situation, then we can avoid the difference of different year. Actually, we think this is a smart method to do the comparison.
Besides, other previous study has also used this method to do the comparison, such as Sun et al. 2017, Effects of green space dynamics on urban heat islands: Mitigation and diversification, Ecosystem Services, 30, 38-46
So we think the method is appropriate to do the comparison.
We hope we have answered your concerns!
Please let me know if you have any questions.
- For the reviewer, the RLST values for Shenzhen are worth noting. Comparison between 1995 and 2015. One can see a significant decrease in RLST despite increasing built-up area. What can such a condition be caused by? Does it not challenge the overall analysis?
Answer: Thank you for your comments!
That is a good question!
As we mentioned, here we used relative LST to do the study, which means: in 1995, Shenzhen has the most significant urbanization process (land cover) while other city of the region is not so significant, so we can see in 1995, Shenzhen is very obviously (it is a relative value!!!). However, in 2005 and 2015, the urbanization of other cities in this region is rapid, which make the Shenzhen is not obvious. For instance, from 2005-2015, Guangzhou-Foshan region experience rapid urbanization process, so we can see this trend in Figure 4.
Actually, we have explained this situation in previous study, you can read it if you are interested:
Zhaowu Yu*, Yawen Yao, Gaoyuan Yang, Xiangrong Wang, Henrik Vejre. Spatiotemporal patterns and characteristics of remotely sensed regional heat island during the rapid urbanization (1990-2015) of Southern China. Science of the Total Environment. 2019, 674:242-254
- Please add to the article (or additional materials) a table showing the calculations for formula (5) for individual parts of the PRDR
Answer: Thank you for your comments!
We have added a table as the additional materials to show the results of each parts of PRDR.
Other necessary fixes:
Maps at work, in particular Figure 3, are of terrible quality and should be improved according to cartographic principles.
Answer: Thank you for your comments!
We have updated the map!
In Figure 5, it should be clearly indicated whether the scale refers to the PRDR map or to the zoom.
Answer: Thank you for your comments!
We have updated the map!
In all tables, consider the accuracy of the numbers. A notation of 0.000 in relation to the% value seems too accurate. In tables 3 and 4, the accuracy of the notation of values is significantly too high. Please round them properly.
Answer: Thank you for your comments!
We have updated the table and round them properly.

Reviewer 3 Report
Reverse thinking: A new method from the graph perspective for evaluating and mitigating regional surface heat islands , Zhaowu Yu et al.,
This paper the authors propose a new method to evaluate and mitigate regional urban heat island (UHI).The method seems correct to me and the results justify their publication, despite the fact, that, they themselves recognised from the outset, a number of limitations of the method itself, seen as a clue for further research, neither presented a method on how to build a heat island network and mitigated heat island. Indeed, they argued that “this study only proposed a (reverse thinking) framework – which is also the most important value of this research, and selected Pearl River Delta Metropolitan Region as the case to verify the validity of the method.”. The method is based on graph perspective which integrates morphological spatial pattern analysis (MSPA). Nevertheless, the method was testes in Pearl River Delta Metropolitan Region and obtained interesting quantitative and qualitative results. They demonstrated how to mitigate UHI, by lowering the temperature as measures to be implemented for sustainability development of those areas.
The paper deals with a very important climate issue, UHI, which is itself an important topic related to regional climate changes associated to the anthropogenic activities. It was easy to read, the English language sounds correct. I, therefore, recommend the publication of this manuscript without any further observation.
Author Response
Thank you for your constructive comments and endorse our study, We also believe this work can contribute quite a lot to related research field.
Reviewer 4 Report
The topic is interesting.
I suggest some minor improvements:
- In the abstract, include more introductory materials and present the identified knowledge gaps.
- A better connection of the method with the issues of urban planning is needed. That is, how this method can be used in the urban planning or urbanization of new areas. It will be useful for these comments to appear in the conclusion
Author Response
Answer: Thank you for your comments!
We have updated the abstract, as well as the sentences connect the method to urban planning.
Please read it.
For instance,
“For example, in this PRDR case, we identified that the core type is becoming more and more important in the process of rapid urbanization, and it could influence the edge type significantly. Hence, this result implies that the cooling measures are crucial for core types to alleviate the expansion of the UHI effect. Which means that when doing the climate adaption planning, it is necessary to consider the construction of sufficient facilities such as green/blue spaces,green roofs, and cool materials in Core type areas to cool down. The results of other types (such as branch and bridge) indicate that these heat patches (or we can call it corridors) are needed to be blocked/destroyed by urban blue-green space and green roof [6,58]. Besides, we quantified the core importance of UHI patches from 1995 to 2015 in PRDR and exampled Zhongshan City, which can provide a direct basis for decision-makers in PRDR to determine the priority of regional thermal environment mitigation and governance. This method is particularly useful in urban renewal based on climate mitigation and adaptation, that is, urban and regional heat island effects can be mitigated by firstly identifying key heat patches and corridors, then block/destroy them. Generally, from a graph perspective, this new method (reverse thinking) integrating MSPA and habitat availability indices can not only determine different UHI types more accurately and take corresponding mitigation measures effectively in the PRDR but can also be widely used in research and practice in other regions.”
Thank you again for your constructive comments!